

# COVID-19 mass media coverage in English and public reactions: a West-East comparison *via* Facebook posts

Ahmad R. Pratama[1,*] and Firman M. Firmansyah[2,*]

[1] Department of Informatics, Universitas Islam Indonesia, Sleman, Daerah Istimewa Yogyakarta, Indonesia

[2] Department of Technology and Society, Stony Brook University, Stony Brook, New York, United States

* These authors contributed equally to this work.

## ABSTRACT

Newspapers and other mass media outlets are critical in shaping public opinion on a variety of contemporary issues, including the COVID-19 pandemic. This study examines how the pandemic is portrayed in the news and how the public reacted differently in the West and East using archival data from Facebook posts about COVID-19 news by English-language mass media between January 2020 and April 2022 ($N = 711{,}646$). Specifically, we employed the Valence Aware Dictionary and sEntiment Reasoner (Vader) to measure the news tone on each COVID-19 news item shared on Facebook by mass media outlets. In addition, we calculated a polarity score based on Facebook special reactions (*i.e.*, love, angry, sad, wow, haha, and care) received by each post to measure public reactions toward it. We discovered that people in Western countries reacted significantly more negatively to COVID-19 news than their East counterparts, despite the fact that the news itself, in aggregate, generally contained a relatively similar level of neutral tone in both West and East media. The implications of these distinctions are discussed in greater detail.

# INTRODUCTION

Newspapers and other forms of mass media play a crucial role in shaping public opinion on a variety of contemporary issues, including the COVID-19 pandemic (*Medina, Rodriguez & Sarmiento, 2021*; *Rozado, 2021*; *Wu, 2021*). Even more so, they not only shaped how people perceive the outbreak (*Motta, Stecula & Farhart, 2020*), but also influenced how people cope with the disease and perform preventive measures against it. For example, in the United States, greater mass media consumption has been associated with a better social distancing practice (*Pedersen & Favero, 2020*) and higher COVID-19 vaccine acceptance rate (*Piltch-Loeb et al., 2021*). Meanwhile in Pakistan, mass media has been one of the reasons why people are otherwise reluctant to get vaccinated (*Ahmad Kamboh, Ittefaq & Sahi, 2022*; *Chaudhary et al., 2021*). In this regard, their journalism styles that commonly employ striking headlines with visual appeal arguably have nudged

Corresponding authors
Ahmad R. Pratama,
ahmad.rafie@uii.ac.id
Firman M. Firmansyah,
manda.firmansyah@stonybrook.edu

Pakistani readers to generalize several cases of reported vaccine adverse side-effects, which subsequently make them against it (*Chaudhary et al., 2021*).

As the contrasting examples above suggest, it appears that the way in which the COVID-19 pandemic is portrayed in the news determines whether the media help or hinder the mitigation efforts. Indeed, even in the United States, where mass media in general helped prevent the spread of the virus as previously mentioned (*Pedersen & Favero, 2020*; *Piltch-Loeb et al., 2021*), some right-leaning media indirectly contributed to its spread by disseminating or even amplifying COVID-19 misinformation (*Motta, Stecula & Farhart, 2020*). Further analysis confirmed that their frequent readers tended to hold false beliefs about the pandemics and to disregard preventive measures such as social distancing (*Motta, Stecula & Farhart, 2020*; *Pedersen & Favero, 2020*; *Zhao et al., 2020*).

Considering the double-edged sword effect mass media can generate, it is important to examine how different media portray the outbreak differently. While past studies have initiated such efforts, they mostly focused on either certain news outlets *e.g.*, (*Hossain, Wahab & Khan, 2022*; *Ogbodo et al., 2020*), specific locations sharing similar cultural backgrounds *e.g.*, (*Mach et al., 2021*; *Rianto & Pratama, 2021*), or relatively small number of different countries (*Wu, 2021*). It is arguably difficult to infer a larger direct comparison, such as the differences between West and East mass media portrayals of the pandemic, let alone examine public reactions to such portrayals, due to these scopes. This study aims to fill this gap in the literature. Specifically, we intend to address the following research questions:

1. How have the mass media covered the COVID-19 pandemic in the past 2 years? Were there any differences in the tone used by mass media in the West compared to the East in their COVID-19 news coverage?
2. Did the audience in the West and East react differently toward COVID-19 pandemic news shared by mass media outlets?
3. Is there any relationship between the COVID-19 news tone by mass media and the public reactions it received?

To answer those questions, we analyzed archival data of Facebook posts by mass media in 18 countries between February 15, 2020, and April 14, 2022, along with their public reactions. Prior to getting into further details, let us highlight some differences between West and East societies in responding to the pandemic that can be explained through the individualism-collectivism cultural dimension (*Hofstede, Hofstede & Minkov, 2005*) along with socio-economic-political contexts and why Facebook posts and special reactions can help us uncover the sentiment behind them in the following sections.

## COVID-19 and individualism-collectivism cultural framework

Individualism-collectivism framework is among the cultural characteristics that distinguish Western civilizations from their Eastern counterparts (*Hofstede, Hofstede & Minkov, 2005*). Individualism culture is characterized by a desire for a loosely knit social structure in which people are expected to care exclusively for themselves and their

immediate families (*Hofstede, 2011*). In contrast, collectivism expresses a preference for a tightly knit social structure in which individuals may expect their family or members of a certain ingroup to care for them in exchange for unquestioned allegiance (*Hofstede, 2011*). In simpler words, individualism culture emphasizes "I" and "me" while collectivism culture emphasizes "we" and "us". While these distinctions hold true at the group level, it may not always be the case at the individual level (*Hofstede, Hofstede & Minkov, 2005*). In this respect, individuals of the same nation, culture, or civilization may have different levels of individualism/collectivism themselves (*Hofstede & McCrae, 2004*), which are then referred to as individualism/collectivism personality traits (*Burton et al., 2021*; *Triandis, 2001*).

Recent studies revealed that the individualism-collectivism cultural differences also manifested in the different responses between people in Western and Eastern countries toward the COVID-19 mitigation policies and prevention efforts. For instance, *Chen, Frey & Presidente (2021)* analyzed daily data on geographic mobility in 111 countries during the early epidemic and discovered that people were less likely to adhere to lockdown regulations in countries with a higher degree of individualism such as the United States, compared to countries with a higher degree of collectivism such as China. Furthermore, *Leonhardt & Pezzuti (2022)* surveyed 400,000 people in more than 50 countries and found that people from collectivistic countries tended to have positive attitudes towards COVID-19 vaccine compared to those from individualistic countries. Connecting the dots, it is less surprising that countries with a greater degree of individualism had higher numbers of confirmed cases and mortality rates than countries with a greater degree of collectivism (*Chen & Biswas, 2022*; *Maaravi et al., 2021*; *Rajkumar, 2021*; *Ritchie et al., 2020*).

## COVID-19 and socio-economic-political factors

Indeed, such a cultural framework is insufficient to explain why some countries in certain regions have higher COVID-19 confirmed cases and mortality rates than others. Other aspects such as socio-economic-political factors seem also to play roles. In terms of confirmed cases, for instance, the rate in a given country has been arguably contingent on the number of available testing kits. In Western countries such as the United States, COVID-19 tests were not only more available but also free (*Assistant Secretary for Health, 2020*). This condition helped authorities track and record such cases. Meanwhile, in Eastern countries such as Indonesia, people had to pay out of pocket to get tested (*Massola, 2020*). This condition arguably not only hindered Indonesians from getting tested but also made it harder to estimate the real number of positive cases in the country. It is not impossible that Indonesia had the same rate of positive cases as the United States, or even greater. Yet, it is difficult to confirm since the testing tools were inadequate in the first place.

The scarcity not only applies to COVID-19 testing kits but also to COVID-19 vaccines. Interestingly, when countries in both regions have arguably similar resources, vaccination rates in the Eastern countries such as Singapore tended to be higher than those in Western countries such as the United Kingdom (*Ritchie et al., 2020*). While it can be attributed to the cultural framework as previously mentioned, *i.e.*, (*Leonhardt & Pezzuti, 2022*), it could

also be due to different political situations. In the Western countries which are predominantly democratic, the governments have less authority to force people to get vaccinated. In contrast, in the Eastern countries, in which the governments are arguably more authoritative despite some of them being democratic (*e.g.*, Singapore), it is not difficult to impose such rules that forcely nudge their citizens to get vaccinated (*Dyer, 2021*).

## Sentiment analysis through facebook posts and special reactions

In comparison to traditional approaches, big data and social media footprints make it easier to undertake nationwide or worldwide sentiment research. In addition, digital footprints on social media capture things as they have occurred, which is not always the case in traditional approaches like surveys and interviews. As a result, these digital footprints have also been extensively employed to study a wide range of social topics on the internet and in particular on social media over the past decade, including those that are quite sensitive such as anonymous online donations (*Firmansyah & Pratama, 2021*), racial digital identities (*Firmansyah & Jones, 2019*), and political preferences (*Oliveira, Bermejo & dos Santos, 2017*).

Apart from being, still, the largest social media in the world as of 2022 (*Statista, 2022*), Facebook also offers a unique way of conducting sentiment analysis studies using its special reactions, as illustrated in Fig. 1. While the reactions themselves in general are intriguing to study (*e.g.*, females, older, less educated users tend to give more 'likes' – *Hong, Chen & Li, 2017*), the special reactions (*i.e.*, love, wow, haha, sad, angry, and care) which were introduced in 2016, have been used to complement text-based sentiment analysis by researchers all around the globe (*Freeman, Alhoori & Shahzad, 2020*; *Oueslati, Khalil & Ounelli, 2018*; *Pratama, 2022*; *Tian et al., 2017*). As a result, it is possible to analyze both the content of the Facebook posts using text-based tools like the Valence Aware Dictionary for sEntiment Reasoning (Vader) (*Hutto & Gilbert, 2014*) as well as how the public reacted to those posts based on the special reactions they received.

# MATERIALS AND METHODS

## Data source

We used CrowdTangle (https://www.crowdtangle.com/), a free public insight tool owned and operated by Facebook to collect our dataset in this study (*CrowdTangle Team, 2019*). This tool allows verified researchers to query databases of more than seven million publicly available Facebook pages, groups, and profiles and their public posts. It is worth highlighting that Meta, the parent company of Facebook, does not allow researchers to perform any data scraping on any of their products even if the researchers aim for a greater good and academic purposes (*Clark, 2021*; *Vittorio, 2021*).

## Data collection procedures

We used the search feature on CrowdTangle using the following parameters: Keyword = covid-19; Platform = Facebook; Account Type = Pages; Timeframe = March 1, 2020– February 28, 2022; Post Type = Links; Language = English, List = Asia Media, Australia

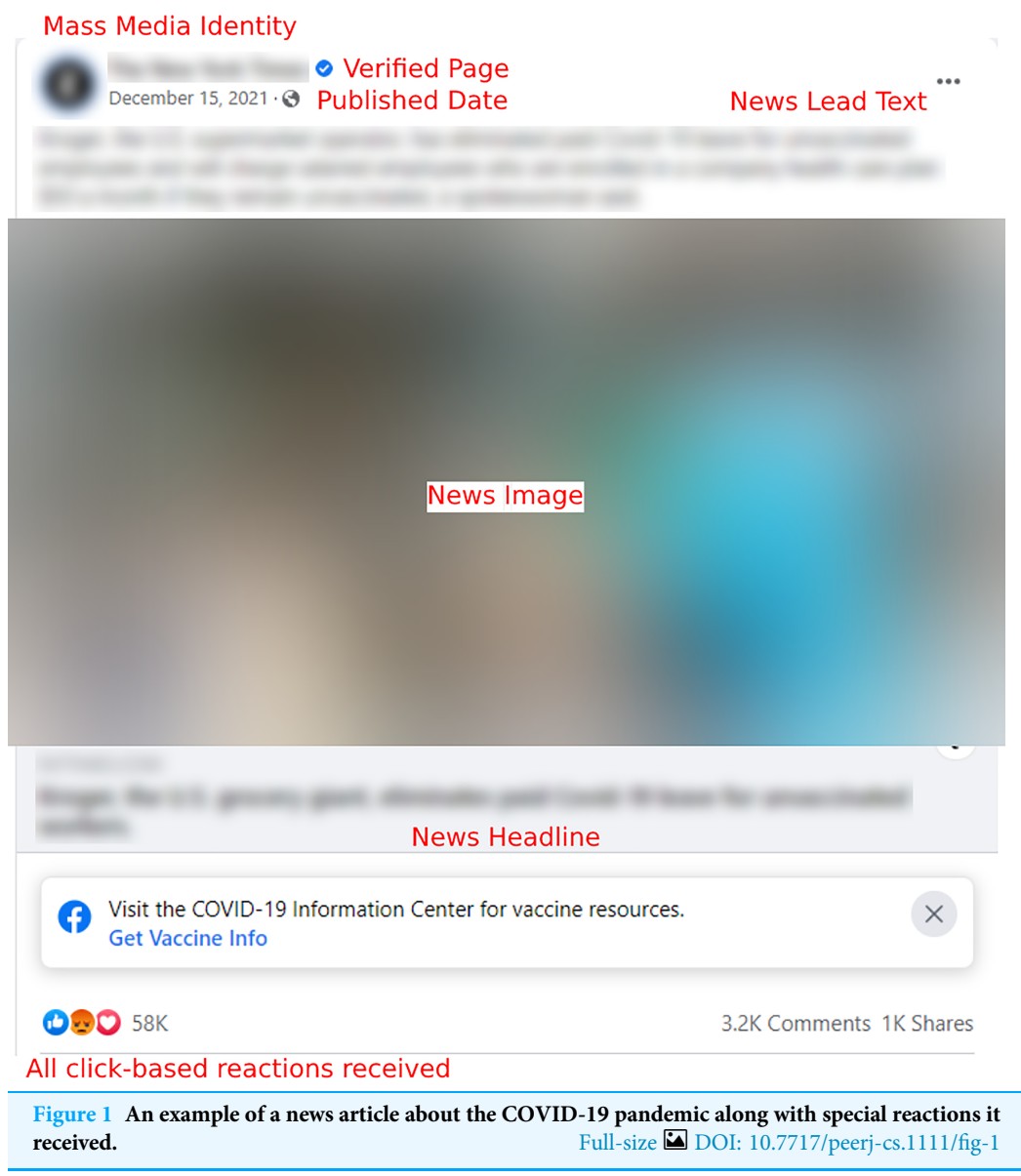

**Figure 1** An example of a news article about the COVID-19 pandemic along with special reactions it received.

Media, Canada Media, UK Media, and US General Media (all managed and maintained by the CrowdTangle team) while keeping any other default settings. This query resulted in a total of 1,003,332 Facebook posts that received a total of 506,401,516 interactions by Facebook users. Each post is essentially a news article shared on Facebook by the respective mass media that links it to the complete news article on their own websites.

In the raw dataset, there were 25 countries in total based on the "Page Admin Top Country" column, some of which had only a small number of posts within the timeframe. Upon further inspection, we decided to omit several countries either due to very limited number of posts such as Denmark (4), South Korea (3), and Taiwan (3) or because their geographical location is not a clear cut to fall within the West or East category, namely Botswana, Israel, Qatar, and Russia. As a result, our final dataset, as summarized in

**Table 1 Number of mass media's Facebook pages, posts, and their public reactions by region, subregion, and country.** Individualism scores were retrieved from *Hofstede, Hofstede & Minkov (2005)*, which did not include scores for all countries.

| Region | Subregion | Country | Number of pages | Number of posts | Total interactions | Total special reactions | Individualism score |
|--------|-----------|---------|-----------------|-----------------|--------------------|------------------------|---------------------|
| West | Europe | Ireland | 11 | 2,951 | 234,932 | 35,678 | 70 |
| | | Isle of Man | 1 | 543 | 39,862 | 10,548 | NA |
| | | United Kingdom | 327 | 109,773 | 47,447,149 | 12,032,252 | 89 |
| | | Subtotal | 339 | 113,267 | 47,721,943 | 12,078,478 | *Mean = 79.5* |
| | North America | Canada | 62 | 208,459 | 62,402,186 | 16,461,408 | 80 |
| | | United States | 69 | 114,417 | 239,334,047 | 81,016,163 | 91 |
| | | Subtotal | 131 | 322,875 | 301,736,233 | 97,477,571 | *Mean = 85.5* |
| | Oceania | Australia | 41 | 50,686 | 23,995,733 | 6,911,059 | 90 |
| | Total | | 511 | 486,828 | 373,453,909 | 116,467,108 | *Mean = 84.0* |
| East | East Asia | China | 2 | 1,201 | 1,446,111 | 38,548 | 20 |
| | Southeast Asia | Cambodia | 1 | 91 | 10,666 | 593 | NA |
| | | Indonesia | 2 | 76 | 5,480 | 1,168 | 14 |
| | | Myanmar | 3 | 1,345 | 260,852 | 7,350 | NA |
| | | Philippines | 20 | 95,559 | 71,455,137 | 22,068,064 | 32 |
| | | Singapore | 14 | 46,674 | 29,045,912 | 6,099,392 | 20 |
| | | Thailand | 16 | 10,221 | 1,189,140 | 241,527 | 20 |
| | | Subtotal | 46 | 153,966 | 101,967,187 | 28,418,094 | *Mean = 21.5* |
| | South Asia | Bangladesh | 4 | 69 | 62,305 | 4,441 | 20 |
| | | India | 41 | 67,709 | 22,091,258 | 3,721,538 | 48 |
| | | Nepal | 5 | 1,526 | 140,720 | 20,402 | 30 |
| | | Pakistan | 6 | 304 | 108,077 | 8,583 | 14 |
| | | Sri Lanka | 3 | 43 | 27,842 | 3,175 | 35 |
| | | Subtotal | 59 | 69,651 | 22,430,202 | 3,758,139 | *Mean = 29.4* |
| | Total | | 107 | 224,818 | 125,843,500 | 32,214,781 | *Mean = 25.3* |

Table 1, consists of 711,646 Facebook posts from 618 mass media in 18 countries, receiving a total of 499,297,409 interactions, 148,681,889 of which were special reactions.

As supplementary information, we added individualism scores for each country from Hofstede's dataset (*Hofstede, Hofstede & Minkov, 2005*) in Table 1. The individualism score ranges from 0 to 100, with higher values indicating higher individualism and in turn lower collectivism, and vice versa. As can be seen, countries in the West have relatively higher scores of individualism ($M$ = 84.0, SD = 8.97) compared to countries in the East ($M$ = 25.3, $SD$ = 10.77). We also added cumulative confirmed COVID-19 cases and deaths per million people as well as vaccination rate for each country included in this study as presented in Figs. 2–4 respectively (*Ritchie et al., 2020*). It is worth noting that the terms "West" and "East" in this paper refer to civilizations and cultures rather than geographical locations. Thus, even though Australia is located in the east longitude and the southern hemisphere, it is still considered as part of the West.

## Data analysis

We analyzed the data in three stages. The first stage serves as a preliminary which anchors further analyses. The second stage aims to answer the first two research questions. The

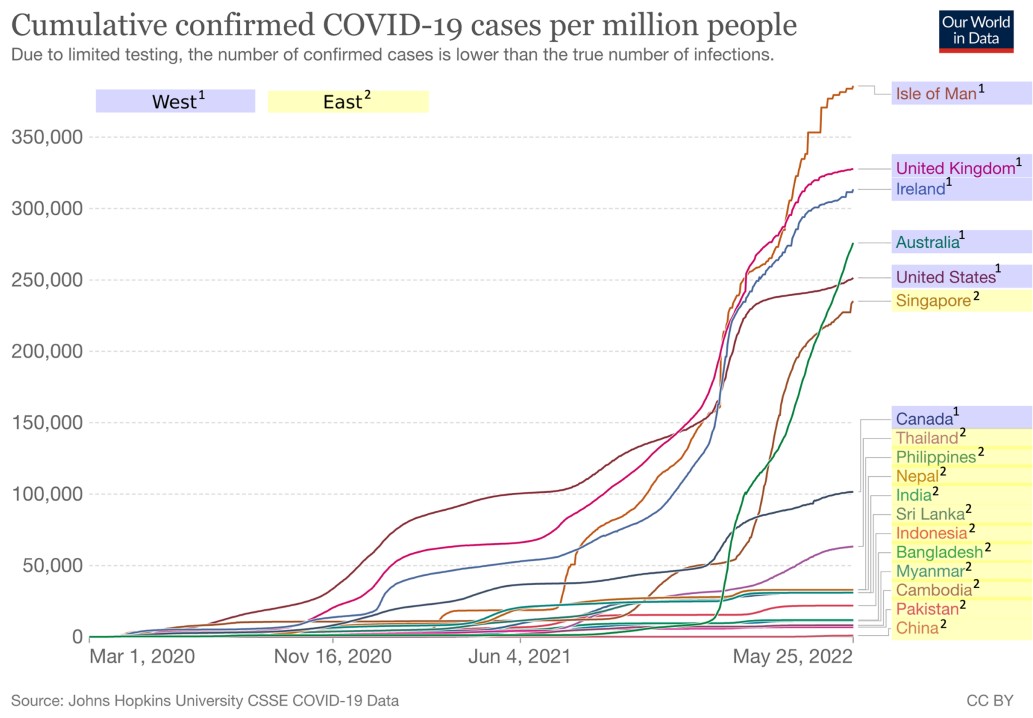

**Figure 2 Cumulative confirmed COVID-19 cases per million people between West and East.** Each line represents a country in either West (blue background) or East (yellow background). Figure source: Our World in Data, CC-BY 4.0, https://ourworldindata.org/explorers/coronavirus-data-explorer.

third stage aims to answer the last research question. All the analyses were conducted using Python on Google Colab. The codes are available on the authors' GitHub repository (https://github.com/ahmadrafie/fbcovid19newssentiment).

### First stage

In the first stage, we applied a text analysis counting the frequency of words in the news then visualizing the results using word clouds with the help of the "wordcloud" package (*Oesper et al., 2011*). In addition to the default stop words (*e.g.*, "and", "or", "only") and punctuations (*e.g.*, ".", ";", "?"), we excluded some other words (*e.g.*, "covid", "covid-19", "coronavirus") and country/city names (*e.g.*, "India", "Canada", "Manila") from the analysis. In this stage, we identified four relevant keywords that were used to subset the dataset for the subsequent specific-keyword analysis. As the subsetting process was mainly based on the identified keywords used in the news, the same news article could be analyzed more than once with different keywords. The subsetting process also made it possible for some articles to be included only in the general analysis (using all dataset) while being excluded from every single specific-keyword analysis (datasets with specific keywords). In other words, the subsets are not mutually exclusive for all news articles.

### Second stage

In the second stage, we treated our dataset as cross-sectional data. We evaluated the content of the news shared by mass media before making comparisons between the West

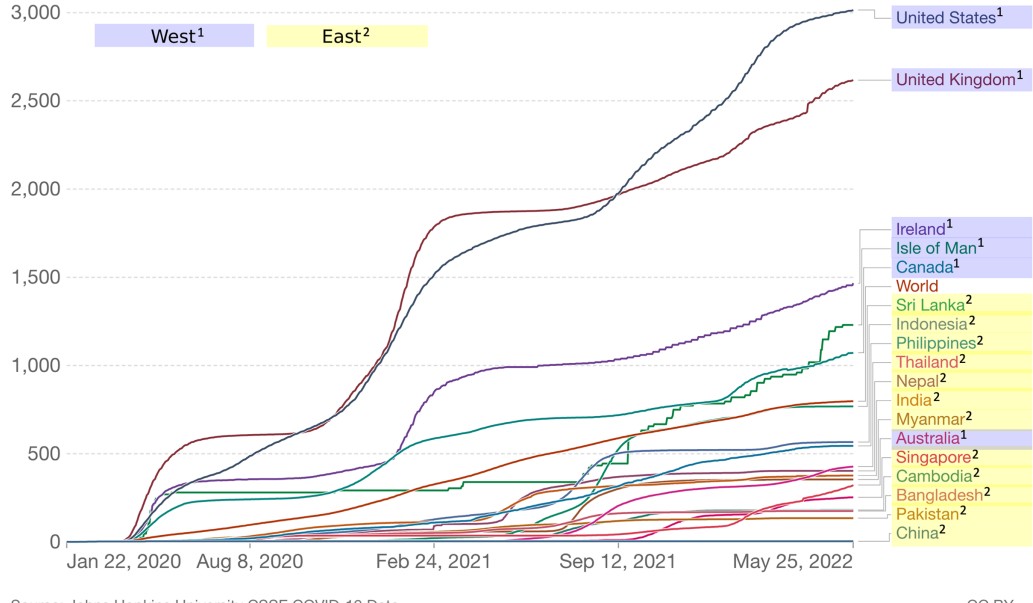

Figure 3 **Cumulative confirmed COVID-19 deaths per million people between West and East.** Each line represents a country in either West (blue background) or East (yellow background). Figure source: Our World in Data, CC-BY 4.0, https://ourworldindata.org/explorers/coronavirus-data-explorer.

and East, both in the tone used in the news and the public reactions it received. We utilized the "vaderSentiment" package on Python (*Hutto & Gilbert, 2014*) to measure the news tone. More specifically, we used the Vader compound score, ranging from −1 for the most negative tone to 1 for the most positive tone for each Facebook post.

As for public reactions, we calculated a polarity score based on special reactions a Facebook post received using the formula specified in the previous studies in the literature (*Freeman, Alhoori & Shahzad, 2020*; *Pratama, 2022*). In doing so, we first utilized the 'love' reaction to indicate a positive valence (1) as opposed to 'sad' and 'anger' reactions for a negative valence (−1). Then we calculated an intensity score, which is a ratio of all special reactions received by a post to all click-based reactions (including "Like"). Finally, a polarity score is computed by multiplying the valence score by the intensity score. Similar to the Vader compound score, the polarity scores also range from −1 for the most negative reactions to 1 for the most positive reactions.

Once we obtained both Vader compound and polarity scores for each Facebook post in our dataset, we then performed a series of Welch's t-tests to examine the differences between news tone and public reactions in the West *vs.* in the East. The Welch's t-test is more appropriate for our dataset given the unequal variances and unequal sample size between the two categories (*Delacre, Lakens & Leys, 2017*). We also computed Cohen's d effect size (*Fritz, Morris & Richler, 2012*) to help interpret the t-test results. We repeated the analysis using smaller subsets of the data by using the top four most relevant keywords

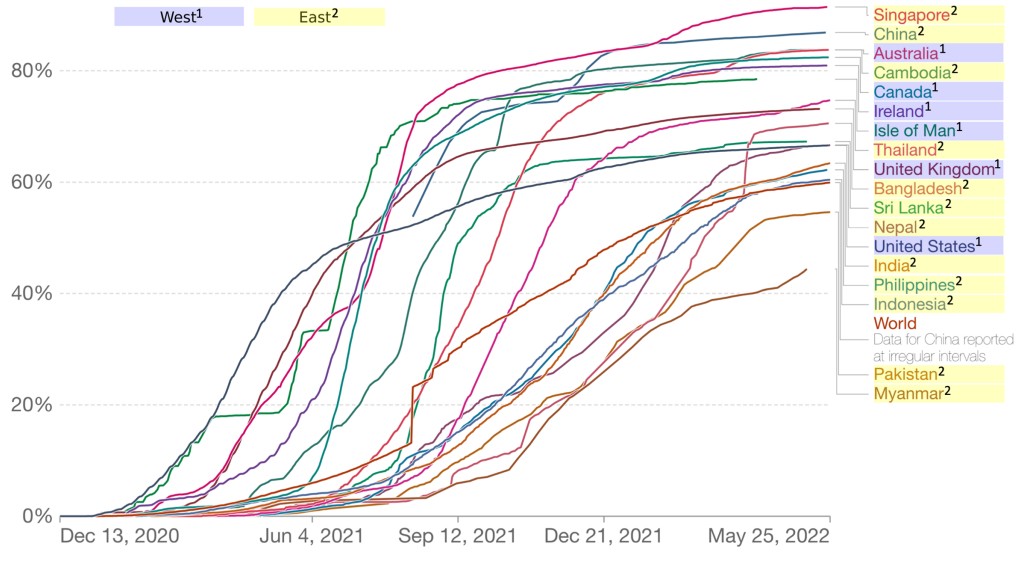

Share of people who completed the initial COVID-19 vaccination protocol
Total number of people who received all doses prescribed by the initial vaccination protocol, divided by the total population of the country.

Source: Official data collated by Our World in Data
Note: Alternative definitions of a full vaccination, e.g. having been infected with SARS-CoV-2 and having 1 dose of a 2-dose protocol, are ignored to maximize comparability between countries.

CC BY

**Figure 4  COVID-19 vaccination rate in West and East.** Each line represents a country in either West (blue background) or East (yellow background). Figure source: Our World in Data, CC-BY 4.0, https://ourworldindata.org/explorers/coronavirus-data-explorer.

in the text to give a more detailed understanding of the differences. This process was done with the help of "Researchpy" package (*Bryant, 2022*).

### Third stage
In the third stage, we aggregated the data by month and region. We visualized the data using line charts and conducted a correlation test between the news tone and the public reactions for both the West and East media. Similar to the second stage, we repeated the analysis using the same smaller subsets of data where we filtered our dataset by four specific-keywords. This process was also done with the help of "Researchpy" package (*Bryant, 2022*).

## RESULTS
### COVID-19 news coverage in the West and East media
Figure 5 shows the word cloud visualizations of the news article posted on Facebook by mass media, either overall or broken down by regions and sentiment of the posts. As previously mentioned, stop words and punctuations were excluded from the preliminary text analysis. Here, we found four relevant keywords, *i.e.*, "case", "vaccine", "health", and "death" for the further specific-keyword analysis to supplement the general analysis based on the overall dataset. It should be noted, we did not consider some other words like "pandemic", "new", or "say" relevant since the former is too similar to the main topic of the

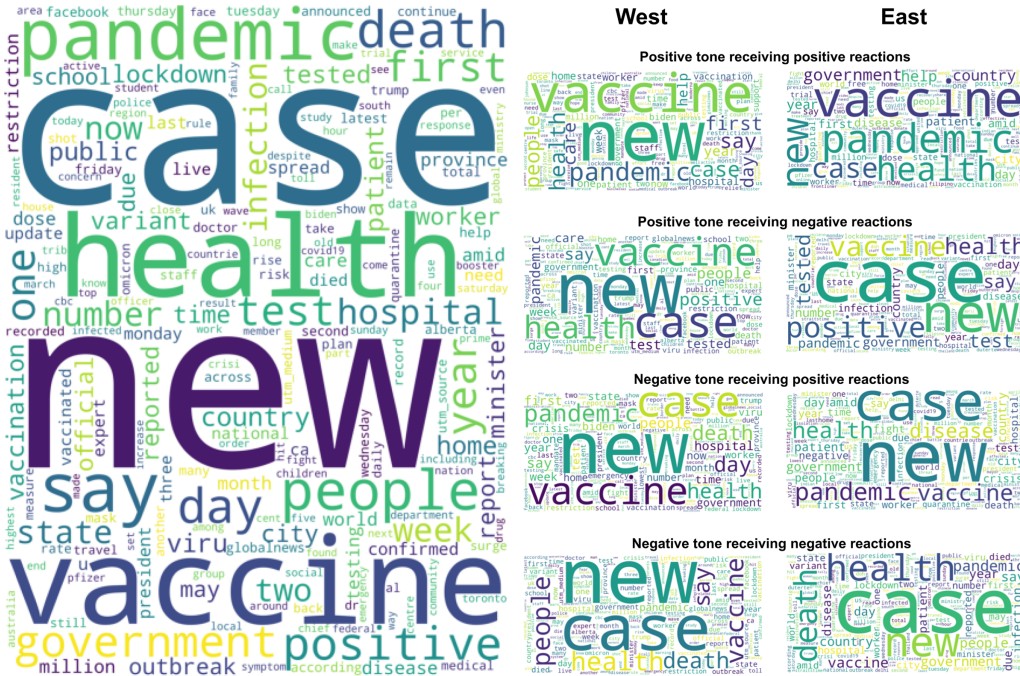

**Figure 5 Word cloud visualizations of COVID-19 news posted on Facebook by mass media.**

overall dataset (*i.e.*, "COVID-19") while the other two are not a noun, making them too general and thus arguably make less sense to be analyzed separately.

Figure 5 (right side) further shows which relevant keywords appear more frequently in which news tone and which public reactions. For example, the word "case" received mostly negative reactions in both regions, but positively toned news containing this keyword is more prevalent in the East than in the West. Another example, the word "vaccine" is mostly covered in a positive tone, receiving more positive than negative public reactions in both regions, but negatively toned news containing this keyword is more common in the West than in the East.

## COVID-19 news tone and public reactions in the West and East

Table 2 shows the percentages of positive and negative public reactions to two different news tones: positive and negative in both regions. As can be seen, in the West region, regardless of the news tone, in general, the public reacted more negatively. This finding applies to all subregions, including Europe, North America, and Oceania. Meanwhile, in the East region, the public reactions show more variations. In aggregate, they reacted more positively only toward positive news tones and more negatively toward negative tones. Interestingly, in the East Asia subregion, the public reacted more positively to all news regardless of its tonality. Even more so, the percentage of those reacting negatively is less than half of those reacting positively.

Furthermore, Table 3 presents the calculated Vader compound scores for the news tone and polarity scores for the public reactions in both the West and East regions, along with

**Table 2  Percentage of positive and negative news tone in mass media's Facebook posts by region, subregion, and public reactions.**

| Region | Subregion | Public reactions | News tone | |
|--------|-----------|------------------|-----------|---|
| | | | Positive | Negative |
| West | Europe | Positive | 38.39% | 21.70% |
| | | Negative | 61.61% | 78.30% |
| | North America | Positive | 35.50% | 22.52% |
| | | Negative | 64.50% | 77.48% |
| | Oceania | Positive | 31.29% | 21.97% |
| | | Negative | 68.71% | 78.03% |
| | | Positive | 35.72% | 22.26% |
| | | Negative | 64.28% | 77.74% |
| East | East Asia | Positive | 82.81% | 73.08% |
| | | Negative | 17.19% | 26.92% |
| | Southeast Asia | Positive | 50.72% | 33.11% |
| | | Negative | 49.28% | 66.89% |
| | South Asia | Positive | 58.59% | 42.72% |
| | | Negative | 41.41% | 57.28% |
| | | Positive | 53.46% | 36.37% |
| | | Negative | 46.54% | 63.63% |

**Table 3  Welch's t-test results comparing news tone and public reactions in the West and East regions for the entire dataset and all four subsets.** Numbers reported in the West and East columns are means with standard deviations within parentheses.

| Dataset | Variable | West | East | $p$-value | Cohen's d | Effect size |
|---------|----------|------|------|-----------|-----------|-------------|
| All | News Tone | −0.003 (0.579) | 0.034 (0.605) | <0.001 | 0.063 | Negligible |
| | Public Reactions | −0.317 (0.418) | −0.106 (0.333) | <0.001 | 0.521 | Medium |
| Case | News Tone | 0.022 (0.511) | −0.011 (0.546) | <0.001 | 0.062 | Negligible |
| | Public Reactions | −0.445 (0.392) | −0.203 (0.289) | <0.001 | 0.668 | Medium |
| Vaccine | News Tone | 0.096 (0.546) | 0.148 (0.553) | <0.001 | 0.094 | Negligible |
| | Public Reactions | −0.202 (0.432) | −0.010 (0.346) | <0.001 | 0.471 | Medium |
| Health | News Tone | 0.005 (0.569) | −0.014 (0.588) | <0.001 | 0.034 | Negligible |
| | Public Reactions | −0.354 (0.431) | −0.154 (0.331) | <0.001 | 0.498 | Medium |
| Death | News Tone | −0.263 (0.555) | −0.374 (0.556) | <0.001 | 0.200 | Small |
| | Public Reactions | −0.531 (0.368) | −0.244 (0.271) | <0.001 | 0.827 | Large |

Welch's t-test results. It should be noted, different from Table 2, Table 3 also presents the results for subsetted datasets based on relevant keywords as previously explained. As can be seen, while all the t-test results indicate significant differences in the news tone between West and East media, only "death" related news has a non-negligible effect size, thus meaningful. In this regard, the mass media in the East portrayed death-related news with a slightly more negative tone than the mass media in the West did. Meanwhile, for all general COVID-19 news, case-related news, vaccine-related news, and health-related news, both West and East media, in aggregate, had a relatively similar neutral tone.

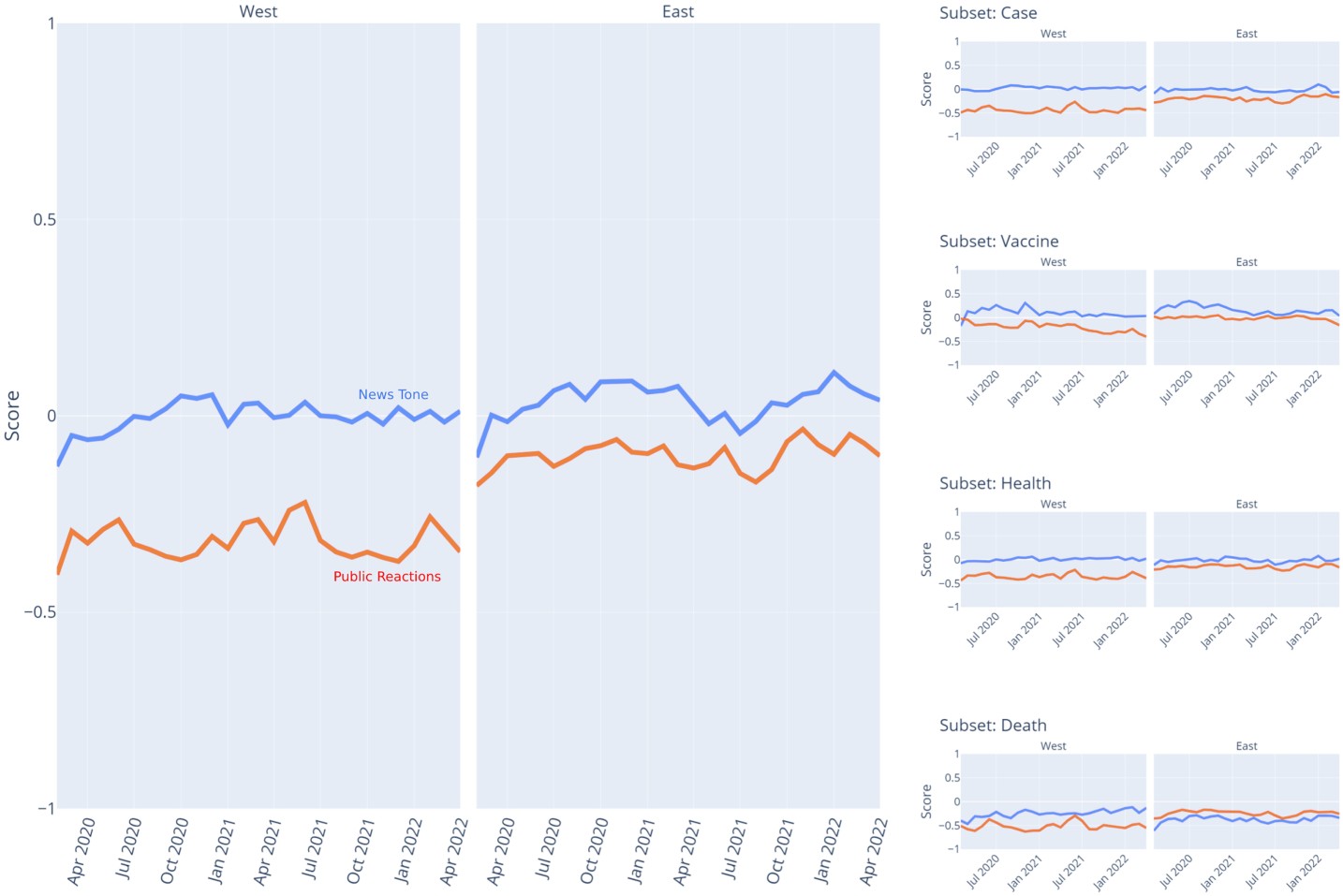

**Figure 6 COVID-19 news tone and public reactions on Facebook over months.** The blue line represents news tone as measured by the Vader compound score, while the orange line represents public reactions as measured by the polarity score based on special reactions.

As for the public reactions, our analysis results indicate that the overall COVID-19 news in the West significantly received more negative reactions than COVID-19 news in the East by the public. Interestingly, these meaningful differences also apply to all specific keywords (*i.e.*, "case", "vaccine", "health", "death") as indicated by medium to large Cohen's d effect sizes.

Moreover, Figs. 6 and 7 present the time series analysis results for the Vader compound scores and public reactions' polarity scores for the general dataset and specific related keywords as well as by subregions, respectively. The visualizations highlight the fact that, aside from the previously mentioned death-related news and the East Asia subregion, the news tones were not significantly different between the mass media in the West and the East. On the other hand, when it comes to public reactions, the COVID-19 news in the West received more negative public reactions than similar news in the East, respectively.

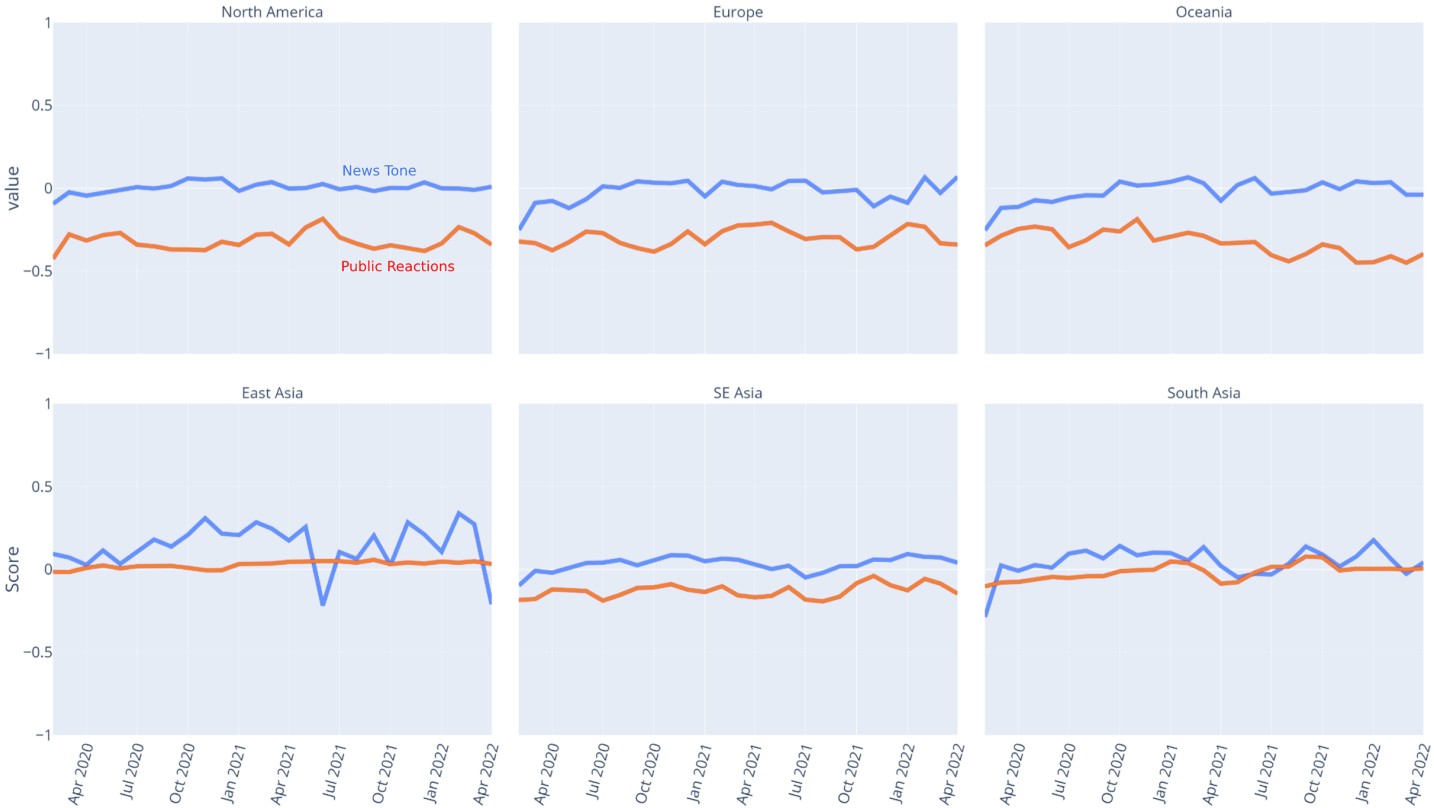

**Figure 7 COVID-19 news tone and public reactions on Facebook over months across subregions.** The blue line represents news tone as measured by the Vader compound score, while the orange line represents public reactions as measured by the polarity score based on special reactions.

**Table 4 Correlation test results between news tone and public reactions (monthly aggregate) in the West and East regions for the entire dataset and all four subsets.**

| Dataset | Region | r value | p-value | Correlation strength |
|---|---|---|---|---|
| All | West | 0.227 | 0.255 | Not significant |
| | East | 0.626 | <0.001 | Strong positive |
| Case | West | −0.291 | 0.142 | Not significant |
| | East | 0.304 | 0.123 | Not significant |
| Vaccine | West | 0.313 | 0.112 | Not significant |
| | East | 0.431 | 0.025 | Moderate positive |
| Health | West | −0.004 | 0.983 | Not significant |
| | East | 0.376 | 0.053 | Not significant |
| Death | West | −0.014 | 0.943 | Not significant |
| | East | 0.596 | 0.001 | Moderate positive |

## Relationships between COVID-19 news tone and public reactions

Table 4 summarizes the correlations between news tones and public reactions in the monthly aggregate. As shown, in general, there is no association between news tones and public reactions in the West. Public reactions in the West are consistently negative

regardless of the tone used by the mass media in COVID-19 news. In contrast, in the East, news tones strongly correlate with public reaction for all datasets in a positive direction. They were also moderately correlated in a positive direction for "vaccine" and "death" related news. In other words, the more positive the tone used by the mass media in the East in news related to the COVID-19 pandemic, including vaccine and death, the more positive the public reacts. Likewise, the more negative the tone used by the mass media in the East in news, the more negative the public reacts.

## DISCUSSION

The analysis results using Vader (*Hutto & Gilbert, 2014*) have revealed that in aggregate, all news articles in English related to the COVID-19 pandemic shared on Facebook by mass media in both West and East regions had relatively similar levels of neutral tone. To be precise, they differed in very minor ways: the East media had a slightly more positive tone (*i.e.*, M = 0.034, SD = 0.605) while the West media had a slightly negative tone (*i.e.*, M = −0.003, SD = 0.579). However, the fact that their difference, albeit statistically significant, has negligible effect size suggests that there is no meaningful difference between them. This significance is mostly driven by the very large sample size in our dataset.

Furthermore, even though some mass media in both West and East regions seemed to intentionally frame the pandemic more negatively as the literature suggests (*Ahmad Kamboh, Ittefaq & Sahi, 2022*; *Chaudhary et al., 2021*; *Motta, Stecula & Farhart, 2020*), there are some others that frame it otherwise (*Pedersen & Favero, 2020*; *Piltch-Loeb et al., 2021*). Thus, they cancel each other out and as a result, put the overall tones in a relatively neutral position. In other words, if we put all media in one basket, their news coverage portrays the pandemic with both positive and negative nuances, resulting in a balanced and rather neutral overall tone. The only exception, however, is for death-related news, in which East media used a significantly more negative tone. Indeed, despite death being unambiguously considered by most as negative news and the fact that negative news attracts more readers (*Trussler & Soroka, 2014*), it is still intriguing to investigate why only this particular topic of COVID-19 was framed with a more negative tone by the mass media in the East.

Regardless of the tone used by the media in their coverage of COVID-19, it is still largely up to the public to express either positive or negative reactions. Even though the public in both regions tend to have negative reactions in general, our findings show that the public in the West has significantly more negative reactions than the public in the East, as shown in Table 3. In other words, people in the West region appear to be perceiving the COVID-19 pandemic as more devastating than those in the East region. One possible explanation for this phenomenon is related to the cumulative cases and deaths per million people, as illustrated in Figs. 2 and 3. In this respect, the top five countries with the highest number of cumulative cases and deaths in our dataset were all located in the West region.

Another possible explanation is individualist culture, which appears to influence how the public in the Western region reacted to COVID-19 security measures and policies (*Chen, Frey & Presidente, 2021*; *Leonhardt & Pezzuti, 2022*). The pandemic has urged people, regardless of their health status, to take various preventive measures to stop the

virus' spread and mitigate casualties. Perhaps these policies have been perceived as threats to freedom and liberty by a sizable portion of the West's population, who predominantly hold individualistic values and whose compliance is primarily motivated by self-interest and a sense of individual obligation to support authorities (*Banker & Park, 2020*; *Favero & Pedersen, 2020*; *Murphy et al., 2020*). As such, they might find preventive measures such as social distancing, mask-wearing, or vaccinations as annoying, inconvenient, or even a threat to their personal freedom (*Johnston, 2022*; *Rains et al., 2022*; *Young et al., 2022*).

Meanwhile, for many people in the East, who predominantly have collectivistic values (*Hofstede, 2011*; *Hofstede, Hofstede & Minkov, 2005*), the said policies might have been perceived as ways to care for each other in their community. They do not mind sacrificing their individual convenience for the sake of their community, so that there is no difference between their self-interest and prosocial motivations behind their compliance (*Chua et al., 2021*). It can also explain why the public in the East showed a more positive reaction to COVID-19 vaccine-related news compared to the public in the West (see Fig. 6). Vaccines not only protect oneself but also others who are even more vulnerable. Indeed, this more positive reaction does not necessarily translate to a higher vaccination rate, as shown in Fig. 4. However, it is arguably due to socio-economic factors that most Eastern countries in this study do not have the same access to COVID-19 vaccines as their Western counterparts (*Acharya, Ghimire & Subramanya, 2021*). Evidently, when they have the means, countries in the East, such as Singapore and China, were among the top three countries in our dataset with the highest vaccination rates. Indeed, such political factors cannot be excluded from the equation. It is also possible that the public in the East region was afraid of the negative consequences imposed by their governments *e.g.*, (*Dyer, 2021*) should they decide to not get vaccinated. It could also prevent or even stop them from making negative comments on social media about COVID-19 news.

In addition, considering that Facebook users with certain demographic backgrounds (*i.e.*, female, older, less educated) tend to give more 'likes' than others (*Hong, Chen & Li, 2017*), it could be the case that users with certain demographics were also more likely to give certain special reactions to the same news article compared to users of other demographics. Perhaps users belong to the former group are more familiar with the Facebook features, have more time to spend on the platform, or possess more means to access the site. Meanwhile, users belong to the latter group tend not to give any special reactions, probably, because they are less knowledgeable of doing it, have less time to spend on Facebook, or deliberately choose not to leave any digital trace on the platform. Since we do not have such demographics data and are restricted to acquire ones by Facebook and CrowdTangle, we could not confirm if our proposed explanation holds in this context.

## Study limitations

With the strengths of using a larger scale of data and employing an innovative method of computational social science, there are some limitations of this study that are worth mentioning. We thus suggest readers be aware of the following limitations when interpreting such results.

First, we only focus on news articles written in English on Facebook with no other topics as a sentiment baseline. We could not include other languages since they have not been supported by the current package used for the main analysis *i.e.*, "vaderSentiment" (*Hutto & Gilbert, 2014*). We were less able to include another topic as the baseline due to our current resources from CrowdTangle prevents us from doing so. This limitation then narrows the countries included in this study and makes Vader, to some extent, as the baseline tool. For the West region, all of them are English speaking countries. For the East region, only a few of them use English as one of their official languages. This limitation also explains why the media in the West region had more Facebook posts and received more special reactions than media in the East. In addition, it is not a given that the news delivered in English in these non-English-speaking Eastern countries shares the same tone or receives the same public reactions as most of the news delivered in their native languages.

Second, we do not have users' demographic information that can help explain the disparity in the public reactions between West and East regions in much greater detail. For one, the information was not provided by CrowdTangle. Indeed, we could do data scraping and additional procedures on our own to get the demographic data. However, it would be against Meta's and Facebook's policies, which forbid data mining in any form, including for academic purposes (*Clark, 2021*; *Vittorio, 2021*).

Third, we only examined associations between variables of interest. Thus, such significant findings or larger effect sizes should not be interpreted as causality. This limitation also applies to the explanation regarding the disparity between public reactions in the West and East. Even though our arguments are supported by past research, we have yet to validate them empirically, for instance, by using statistical procedures.

Finally, the facts that Facebook penetrations were not the same across the countries analyzed in this study (*Dixon, 2022*) and different mass media might use Facebook differently give a further precaution for readers when making such generalizations. In this regard, the results, albeit derived from a relatively huge dataset, might not necessarily reflect the whole population in the country under study. Likewise, the results may not always be representative of the entire media system in the regions under study. Some mass media might use Facebook more frequently and thus have a greater chance to be in our dataset than other media. Unfortunately, this current study was less able to give detailed pictures of the types of media that were specifically examined, except the fact that they were all digital mass media sharing news articles both in their own website and Facebook.

## Directions for future research

Software-wise, we encourage future developers to address the current tool's limitations by, for instance, developing sentiment analysis packages in other languages that are compatible with Vader to allow for meaningful comparison with its English counterpart. In addition, we recommend future studies to use other points of views in examining the differences or similarities between news tones and public reactions in the West and East regions. For instance, it would be interesting to investigate the comparisons through political leanings of the media or the readers. It would also be more fruitful to learn

demographic and examine other psychological factors including individualism/collectivism personality traits that may influence such reactions. Finally, we also motivate future researchers to do a replication study, in particular with smaller units of analysis (*e.g.*, country level) or by using a sentiment baseline. Doing so, we believe, will further validate the study findings.

## CONCLUSION

This study aims to investigate COVID-19 news sentiment by looking into the differences in tone used by the mass media in the West and East in their COVID-19 pandemic news coverage in English, as well as public reactions to it, and if there is any correlation between the two. Through analyzing archival data of news posted by mass media on Facebook with the help of a sentiment analysis software, Vader, we discovered that, in aggregate, news in both West and East media generally contained a relatively similar level of rather neutral tone. However, people in Western countries reacted significantly more negatively to COVID-19 news than their Eastern counterparts. Interestingly, we found none to strong correlations between the news tones and public reactions in both regions suggesting case-by-case relationships depending on the topics. We then argue that this disparity might be driven by the individualism *vs.* collectivism cultural differences along with different socio-economic-political contexts between countries in the West and East regions.

### Funding
The authors received no funding for this work. The funders had no role in study design, data collection and analysis, decision to publish, or preparation of the manuscript.

### Competing Interests
The authors declare that they have no competing interests.

### Author Contributions
- Ahmad R. Pratama conceived and designed the experiments, performed the experiments, analyzed the data, performed the computation work, prepared figures and/or tables, authored or reviewed drafts of the article, and approved the final draft.
- Firman M. Firmansyah conceived and designed the experiments, performed the experiments, analyzed the data, performed the computation work, prepared figures and/or tables, authored or reviewed drafts of the article, and approved the final draft.

### Data Availability
The summarized and anonymized data along with the code used to processed it are available at GitHub: https://github.com/ahmadrafie/fbcovid19newssentiment.

Per CrowdTangle's policy, the raw dataset cannot be shared publicly.

The dataset can be retrieved by a CrowdTangle user with the following search parameters: Keyword = covid-19; Platform = Facebook; Account Type = Pages; Timeframe

= March 1, 2020 - February 28, 2022; Post Type = Links; Language = English, List = Asia Media, Australia Media, Canada Media, UK Media, and US General Media (all managed and maintained by the CrowdTangle team) with all other default settings.

## Supplemental Information

Supplemental information for this article can be found online at http://dx.doi.org/10.7717/peerj-cs.1111#supplemental-information.

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
