# Peer review of "COVID-19 mass media coverage in English and public reactions: a West-East comparison via Facebook posts"

_PeerJ Computer Science, doi:10.7717/peerj-cs.1111_

## Round 0.1 · original submission · Major Revisions

Both reviewers value the contributions of this work, but believe that major revisions are needed before the work can be considered again for review. Most importantly, reviewers highlight a number of aspects that could be improved in terms clarity of report, detail and depth of the analysis.

Reviewer 2 also provides additional references to be considered in fleshing out the discussion of, and distinction from, related work.

As well as the reviewers, I would also like to hear a clarification on the availability of the data used in this study, which at the moment seems to be limited to only part of the data.

Reviewer 1 ·

Basic reporting

This study investigated COVID-19 news sentiment by looking into the differences in news tone posted by mass media on Facebook and public reactions to news in the West and East regions. The aim is interesting, the manuscript is well-written, and the data and code are available. However, I have some comments to be improved.

1. Captions for figures and tables should be more descriptive.
2. Table 1: Values in the column of Number of Posts should be separated by a comma, not a period. In addition, why the individual score in some countries is NA?
3. For easy comparison, countries in West and East in Figures 2 and 3 should be distinguished similar to Figure 4.
4. The order of Figures 5 and 6 are reversed. The caption at the top of the figure is not needed because it is double-captioned in Figures 5 and 6.
5. As the authors mentioned in the limitations (Lines 322-324), news tone and readers would be different in English news and native language news in non-English-speaking Eastern countries. Considering this, “English” should be appeared in the paper title, e.g., “Facebook posts in English”.
6. I recommend the authors organize the Discussion section by creating subsections, etc. The Limitation s and Directions for Future Research section would be moved to the Discussion section. Since three research questions are listed in the Introduction, it would be great if the authors clearly answer the questions in the Results or Discussion sections.

Experimental design

1. Although a polarity score based on special reactions was calculated as public reactions in the Second stage, weren’t other “interactions” used? I agree special reactions are more informative but the total number of special reactions is very small and many interactions are ignored according to Table 1. What do you think is there any bias or influence from this?

Validity of the findings

1. I couldn’t understand Welch’s t-test results in Table 2. What are the values in columns “West” and “East” compared to what? Also, what are the values in parentheses? Please give more detailed explanation for Table 2.
2. Lines 220-221: Although the authors describe that “While all the t-tests results indicate significant differences of the news tone between West and East media, only those about “vaccine” and “death” that are meaningful”, I don't know why it could be interpreted this way from the results in Table 2. Also, I think the interpretation should be discussed in the Discussion section, not the Results section.

Reviewer 2 ·

Basic reporting

The paper compares the mass media sentiment for COVID-19-related news in English between Eastern and Western countries on Facebook. It also aims to contrast the public reaction to these news posts in the two hemispheres. Finally, the authors investigate the correlation between the news sentiment and the users' average response.

In general, the paper is well written. The language is clear and easy to follow throughout the document. The document is also well organized. The Introduction adequately presents the research questions and properly motivates their relevance. In addition, the references included for contextualizing the problem and discussion are relevant. I appreciate the effort made by the authors in addressing this topical subject.

The tables and figures are easy to read. The main content of the document adequately addresses all of them. However, in Figures 5 and 6, the images and labels are crossed.

The authors have not made the raw data used for the study available. Only aggregated data at the country/month-level is shared. The datasets needed to run the code available on Github are not shared. Even though the authors include the parameters used to query the CrowdTangle system, the processing and exclusion criteria applied over the initial dataset are underdescribed in the paper.

Experimental design

In this study, the Individualism-collectivism framework is used as a rationale to understand the difference in behavior between East and West media and the public reaction. Although there is a relatively long section dedicated to this framework (e.g., compared to sentiment analysis), there is no further engagement or research connected to it. It is only used to justify the division of the media into these two regions. Furthermore, this framework may give us some insights, but multiple other socio-economic and political factors could partially explain the observations. The authors could scale back the description of this framework and mention different dimensions that might influence the engagement around this topic.

Similarly, the authors try to contextualize the observed social media behavior with other pandemic-related figures like the number of cases, the number of deaths, and the vaccination rate per county. As with the individualism framework, there is no further analysis linked to these numbers (even when 3 out of 6 images are dedicated to them). Once again, all these aspects are also influenced by socio-economic factors independent of the cultural dimension of collectivism or stance (political or otherwise) toward COVID-19 or vaccination (e.g., access to vaccines, totalitarian regimens, healthcare infrastructure). The difference in access to vaccines between these two sets of countries is briefly mentioned only in the discussion. This should come way earlier in the paper.

Regarding the dataset, the authors mention that some countries were left out. However, there is no detail on the exclusion criteria. For example, did they use an absolute threshold or a sample percentage? Instead, they give some examples of countries not included. This is important for the reproducibility of the dataset (given that only the initial query is shared).

Within the data processing, there is a step where the authors select keywords. However, it is unclear from the document if the selection was based on a visual inspection of the word cloud or directly on the word counts. Without further clarification, the selection seems somewhat arbitrary. For example, they ignore "new", but I suspect that in most cases, it came together with "case" (as in "new cases"). So, one could say "new" is as general as "case".

Validity of the findings

The discussion of the results seems, at times, naive. For example, the authors found that:
Concerning death-related news, "The fact that the media in the East used a significantly more negative tone than those in the West, however, is somewhat puzzling."
However, this fits the same pattern of individualism, i.e., West media tried to minimize the impact of the pandemic and create a less negative public perception (which may influence the vaccination rate and preventive behaviors). Instead, they seem to favor politically biased and economically focused narratives (see, for example, (Zhao et al. 2020)).

Additionally, it is not easy from the current analysis to establish a direct comparison between the two regions. For example, the authors found that:
"Interestingly, even though the public in both regions has negative reactions in general, our findings show that the public in the West has significantly more negative reactions than the public in the East."
As the authors noticed with the individualism framework, a cultural dimension might influence these results. The same collectivism considered before may influence how people express themselves or interact in social networks. This applies not only to the public reaction but also to the mass media. Although news media coverage tends to have always a negative tone (negative news sell more -- (Trussler and Soroka, 2014)), previous studies have shown that different regions/countries have different baselines, which could affect what or how negatively the public perceives a piece of news (see, for example, (Rudra et al. 2021), (Koch et al., 2020)). So, sentiment baselines should be considered here.

In general, the authors present the scope of the analysis at the regional level as a contribution over previous studies that focus on a few countries or sources. However, they do not reflect on the limitations of this approach. Although countries within the same region probably share more similarities than with countries in the other region, generalizing public behavior and media systems in such a bast area is also unrealistic. For example, having countries like China, India, and Pakistan in a single monolithic system is difficult to argue.

Finally, the validity of the data as representative of these two regions is unclear. Despite being the largest Social Network in the world, Facebook penetration is not the same in all regions/countries. For example, according to some statistics, Facebook penetration in Asia is under 25% compared to 82% in North America or (65-67%) in Europe and Australia. Can we generalize the reaction on Facebook as a public perception for East countries without knowing the sample's demographics? Are they directly comparable to Western data?
For example, the Philippines and Singapore account for 80% of the reactions. However, these two countries have just 5 and 109 million populations, respectively. In contrast, India makes up 17% of the interactions and has a population of over 1.3 billion people. Also, this leaves only 3% of the reactions for the other nine countries (including China).
Moreover, even for the Western sample, less than 1/3 of the post come from the USA, while they contribute 2/3 of the reactions. Thus, users' perception analysis will be biased toward one media system that is not proportionally represented in the sample and can hardly represent the region's public opinion.

Zhao E, et al., "Media trust and infection mitigating behaviours during the COVID-19 pandemic in the USA", BMJ Global Health 2020;5:e003323.

Trussler M, Soroka S. Consumer Demand for Cynical and Negative News Frames. The International Journal of Press/Politics. 2014;19(3):360-379. doi:10.1177/1940161214524832

Rudra, K., et al. "My EU= Your EU? Differences in the Perception of European Issues Across Geographic Regions." IEEE Transactions on Computational Social Systems 8.6 (2021): 1475-1488.

Koch, C. M., et al., "Public debate in the media matters: evidence from the European refugee crisis," EPJ Data Science, vol. 9, no. 1, p. 12, 2020.

Additional comments

In the abstract, the authors mention the size of the initial prefiltered sample obtained from CrowdTangle. This is misleading. According to the methodology, the analysis is only based on less than half of that initial sample (483K posts). However, this sentence in the abstract implies that the research and conclusions are based on over a million posts.

Throughout the paper, the authors use China's example several times as a representative from the East side. However, China is not a good example of the vaccination rate or many other metrics as they can impose much stricter regulations at the national level than many western democracies or even neighboring countries. This is, in general, not based on the cultural background but on the totalitarian regime that controls aspects such as distribution of information, total lockdowns, and vaccination policies, among others. All these have a direct impact on the mentioned metrics.

In the Data Analysis section (lines 176-176), the authors reference the subsetting process and keyword identification. However, this process is not described before. At this point in the text, the reference is confusing.

In line 205, there is a reference to the first stage ("Similar to the first stage"). Probably it should say "second" stage.

In line 261, there is an extra "and" --> "except for two topics: and death-related".

---

## Round 0.2 · Minor Revisions

The paper should become acceptable once the final minor -yet important- revisions suggested by R2 are addressed.

Reviewer 1 ·

Basic reporting

The revised manuscript has sufficiently addressed my comments. I now recommend acceptance for publication.

Experimental design

The revised manuscript has sufficiently addressed my comments. I now recommend acceptance for publication.

Validity of the findings

The revised manuscript has sufficiently addressed my comments. I now recommend acceptance for publication.

Additional comments

The revised manuscript has sufficiently addressed my comments. I now recommend acceptance for publication.

Reviewer 2 ·

Basic reporting

The paper compares the mass media sentiment for COVID-19-related news in English between Eastern and Western countries on Facebook. It also aims to contrast the public reaction to these news posts in the two hemispheres. Finally, the authors investigate the correlation between the news sentiment and the users' average response.

The authors have addressed all of the comments from the first review.

An additional comment would be that, in the Data Analysis section, the authors mention they based their public reaction polarity score on a subset of Facebook's special reactions. However, given the importance of this polarity score for the analysis (and that it is not a simple counting), they should add more details of the Reaction Frequency-Inverse Document Frequency (RF-IDF) here to make the document more self-contained. For example, the cited paper (Freeman et al., 2020) also considers the "Like" reaction, and it is not clear if this paper also considers this reaction. Also, is the reweighting/normalization of the different special reactions made at the region level or for the whole dataset? Making the normalization by region (i.e., East and West) would make the results more comparable.

Experimental design

Most of the comments from the previous review were addressed.

Validity of the findings

Some suggestions from the previous review for using sentiment and reaction baselines were included only as limitations but not considered for the current analysis. IMHO, this decision will limit the validity and potential impact of the present study. However, if they are to be included as limitations, these should also be reflected in the scope of the current discussion.

For example:
The authors mention the language and demographic limitations and potential cultural differences that may affect the interpretation of the results and conclusions from the analysis. However, these limitations are not reflected in the scope of their own discussion and conclusions. For example, in the Conclusion section, the authors write:
"This study aims to investigate COVID-19 news sentiment by looking into the differences in tone used by the mass media in the West and East in their COVID-19 pandemic news coverage ..."
and
"... people in Western countries reacted significantly more negatively to COVID-19 news than their Eastern counterparts."
These may suggest that the results could be directly generalized to the entire media system or population in these regions. Similarly, the Discussion section lacks references to the exact scope (i.e., Media coverage in English and its Facebook audience).

Additional comments

The authors should rephrase the description around Figure 5 and the selected keywords in the Results section. With the proposed methodology, I think we cannot go as far as to suggest that these keywords are "eliciting" the public reaction. As the authors mention in another section, there is also the factor of the reporting sentiment, the reporting outlet, etc. A language similar to the one used in the discussion would be more appropriate (e.g., "reaction to COVID-19 vaccine-related news").

---

## Round 0.3 · accepted · Accept

I appreciate the authors' effort in addressing the final minor concerns from reviewer 2. The current revision satisfactorily addresses these concerns and therefore I recommend acceptance.